# The Relationship between Occupationally Exposed Arsenic, Cadmium and Lead and Brain Bioelectrical Activity—A Visual and Brainstem Auditory Evoked Potentials Study

**DOI:** 10.3390/brainsci11030350

**Published:** 2021-03-10

**Authors:** Marta Waliszewska-Prosół, Maria Ejma, Paweł Gać, Anna Szymańska-Chabowska, Magdalena Koszewicz, Sławomir Budrewicz, Grzegorz Mazur, Małgorzata Bilińska, Rafał Poręba

**Affiliations:** 1Department of Neurology, Wrocław Medical University, 50-367 Wrocław, Poland; marta.waliszewska@umed.wroc.pl (M.W.-P.); maria.ejma@umed.wroc.pl (M.E.); magdalena.koszewicz@umed.wroc.pl (M.K.); slawomir.budrewicz@umed.wroc.pl (S.B.); malgorzata.bilinska@umed.wroc.pl (M.B.); 2Department of Hygiene, Wroclaw Medical University, 50-367 Wroclaw, Poland; 3Department of Internal Medicine, Occupational Diseases, Hypertension and Clinical Oncology, Wroclaw Medical University, 50-367 Wroclaw, Poland; anna.szymanska@umed.wroc.pl (A.S.-C.); grzegorz.mazur@umed.wroc.pl (G.M.); rafal.poreba@umed.wroc.pl (R.P.)

**Keywords:** occupational exposure, arsenic, cadmium, lead, evoked potentials

## Abstract

The aim of this study was to evaluate the parameters of visual and brainstem auditory evoked potentials in patients occupationally exposed to arsenic, cadmium and lead. The study group comprised 41 copper smelter and refinery workers (average age: 51.27) with occupational exposure to arsenic, cadmium and lead. The control group consisted of 36 healthy volunteers (35 men and 1 woman, aged 27–66, average age: 51.08). Neurological examination, brain imaging, and visual and brainstem auditory evoked potentials were performed, and the relationship between blood Cd, Pb concentration (Cd-B, Pb-B), blood zinc protoporphyrin (ZnPP), and urine As concentration (As-U) were assessed. In the workers, exceedances of allowable biological concentrations were observed, with the urinary concentration of arsenic being 5.2%, the cadmium and lead in blood being 1.3%, while the case of ZnPP was 2.6%. The mean P100, relative P100, and N145 visual evoked potential (VEP) latencies were significantly longer in exposed workers than in the controls. The mean wave III and V brainstem auditory evoked potential (BAEP) latency and the mean wave III–V and I–V interpeak latencies were longer, and the I and V amplitude was lower in the workers than the controls. In summary, occupational exposure to As, Cd, and Pb is associated with prolonged latency and reduced evoked potential amplitude, but As-U, Pb-B, Cd-B, and ZnPP concentrations are not linearly related to potential components. The analysis of evoked potentials may be a useful method of assessment of the central nervous system in patients with occupational exposure to heavy metals.

## 1. Introduction

Disorders resulting from chronic exposure to heavy metals affect various organs, including the nervous system. The chronic neurotoxic effects of heavy metals are revealed in workers who are professionally exposed to them. Permanent exposure in the work environment can lead to absorption by the body and, consequently, to damage to the nervous system [1,2]. Knowledge about the molecular and biochemical mechanisms responsible for the direct action of metals on the structures of the nervous system is still insufficient [3,4]. The pathomechanism of the effect of metals on neurophysiological functions and the bioelectrical activity of the central nervous system (CNS) is also not fully known. Clinical symptoms are non-specific and often poorly expressed. The most common manifestations of CNS damage include encephalopathy, cranial nerve palsy, cerebellar ataxia or extrapyramidal system damage, psychoorganic syndromes, and in extreme cases, epileptic seizures and psychotic disorders [5,6,7].

Arsenic, cadmium, and lead are among the most common toxic metals. Occupational exposure to arsenic (As) occurs during the production of pesticides, herbicides, and other agricultural products. Significant exposure to high concentrations of arsenic smoke and dust may occur in the steel industry [8]. Trivalent As compounds are responsible for toxic effects and pentavalent As compounds are responsible for phosphorylation, oxidation, and decomposition. One characteristic of As is its predisposition to cause lesions within the peripheral nervous system in the form of severe, or less often sensory, neuropathy [1,6,9]. Harmful effects on the CNS, cognitive impairment, the development of depression, and an influence on body coordination have also been demonstrated [2,7].

Occupational exposure to cadmium (Cd) is usually inhaled. Workers involved in the purification of lead and zinc ores, the combustion of fossil fuels, galvanization, cement production, and the production of pigments and nickel-cadmium batteries are particularly vulnerable. Cd neurotoxicity most likely results from the mechanism of induction and the intensification of oxidative stress, in particular the intensification of the lipid peroxidation processes [10,11]. Epidemiological studies show that there is a relationship between exposure to Cd and behavioral changes and lowered intelligence in children and adults [12].

Lead (Pb) is a ubiquitous toxic metal that is present in all elements of the environment and biological systems. Depending on the dose and duration of exposure, Pb can cause many side effects in people [13]. Professionally exposed adults may have signs of damage to the central and peripheral nervous systems. Pb affects all neurotransmitter systems in the brain: glutaminergic, dopaminergic, and cholinergic. Clinical manifestations of intoxication most often take the form of encephalopathy, cognitive impairment, hearing damage, and peripheral axonal-demyelinating nerve neuropathy [3,6,7].

An evoked potentials study is a precise, non-invasive, and very sensitive technique for assessing brain bioelectrical activity. It is especially useful for detecting small, subclinical disturbances in patients without signs of damage to the CNS or with only a slight neurological deficit.

The aim of our study was to evaluate the electrophysiological parameters of visual and brainstem auditory evoked potentials in patients without clinical neurological deficits occupationally exposed to arsenic, cadmium, and lead. The analyzed parameters were correlated with basic toxicological parameters.

## 2. Materials and Methods

A total of 77 people were qualified for this study, including 75 men and 2 women. The study included 41 patients (40 men and 1 woman, aged 27–66, average age: 51.27) with occupational exposure to arsenic, cadmium, and lead, who were workers of the copper smelters and refineries “Legnica” and “Głogów” (Poland). The criteria for inclusion in the group of persons professionally exposed to arsenic, cadmium, and lead were: current employment at workplaces with documented exposure to a steel mill according to the safety and health departments, a total period of work in positions with documented exposure to metals of at least 5 years, and lack of current professional exposure to other chemical and physical substances documented by foundry services responsible for work safety and hygiene.

The control group consisted of 36 healthy volunteers (35 men and 1 woman, aged 27–66, average age: 51.08), appropriately selected in terms of sex and age. Everyone included in the study lived in areas with similar environmental exposure characteristics, so they did not differ in the size and quality of environmental exposure. All patients gave informed written consent to participate in the study.

Exclusion criteria from the study were the same for the control group and the testing group. Exclusion criteria included factors that may affect visual evoked potential (VEP) and brainstem auditory evoked potential (BAEP) parameters, namely: visual and auditory receptor damage, current and/or past neurological, psychiatric, metabolic and deficiency diseases, and taking medications that may affect brain bioelectrical activity (e.g., neuroleptics, antiepileptic drugs, hypnotics, glucocorticosteroids). The general characteristics of the studied groups are presented in Table 1.

All study participants completed a standardized survey to obtain information on employment history (department, position, and duration of exposure), lifestyle-modifying factors (diet type, physical activity, smoking, and alcohol consumption), health status, and medical history.

Approximately 25 mL of venous blood and a standard urine sample were collected from all subjects in the group exposed to lead, cadmium, and arsenic. Laboratory tests determined the levels of blood cadmium (Cd-B), blood lead (Pb-B) blood zinc protoporphyrin (ZnPP), and urine arsenic (As-U). Determination of lead and cadmium in blood was performed by atomic absorption spectroscopy in a graphite furnace with a Solaar M6 apparatus, Thermo Elemental, UK. Arsenic concentration in the urine was determined by atomic absorption spectroscopy using the hydride generation method (HGAAS) with a VP100 Continuous Flow Vapor System. The concentration of ZnPP in blood was determined with a fast fluorimetric method using a Hematofluorymeter ProtoFluor Helena (USA). The determinations were made in accordance with the standards of the Institute of Occupational Medicine in Łódź. More information on the applied methodology for the determination of lead, cadmium, arsenic, and zinc protoporphyrin can be found in our previous work [14]. For the performed laboratory analyses, precision errors and coefficients of variation were determined. The precision errors were calculated according to the formula: “±2 * standard deviation.” The coefficients of variation were calculated according to the formula: “(standard deviation/mean) * 100%.” The obtained values of precision errors and coefficients of variation were within the limits of laboratory standards.

Detailed subjective and objective neurological examination, VEP, and BAEP studies were performed on both groups. Each of the exposed patients underwent head imaging—either computed tomography (CT), or magnetic resonance imaging (MRI). All of them underwent mini-mental state examination (MMSE), clock drawing test (CDT), and symbol digit modality test (SDMT) tests to assess basic cognitive functions.

Evoked potential studies were carried out using the Nicolet Viking Quest apparatus, in accordance with the standards of the International Federation of Clinical Neurophysiology (IFCN) and the American Society of Electroencephalography [15,16]. In the VEP study, the wave latency (or wave) of N75, P100, and N145; the inter-ocular difference in P100 wave latency (relative latency), and the amplitude of the P100/N145 complex were evaluated. In the BAEP study, absolute latencies of waves I, III, and V, the interlatencies I–III, III–V, and I–V, and the amplitudes of the I and V waves were assessed. We used the same methods as in our previous work [17].

The research was approved by the Bioethics Committee of the Medical University in Wroclaw.

Statistical analyses were performed based on the statistical program “STATISTICA 13” (Dell Inc., USA). Arithmetic means (X), medians (Me), and standard deviations (SD) were calculated for the quantitative variables. The distribution of variables was checked by the Shapiro–Wilk W test. For independent quantitative variables with normal distribution, the t-test for unrelated variables was used for further statistical analysis. For variables with a non-normal distribution, the Mann–Whitney U test was used for independent quantitative variables. The results for qualitative (nominal) variables were expressed as percentages. For qualitative variables, the chi-square test of maximum likelihood was used for further statistical analysis. To determine the relationship between the studied variables, a correlation analysis was performed. The results at the level of *p* <0.05 were considered statistically significant.

## 3. Results

### 3.1. Toxicological Parameters

The average concentrations of blood cadmium, blood lead, and urine arsenic in the examined group of workers exposed to metals were 1.19 ± 1.40 μg/L, 181.85 ± 139.88 μg/L, and 15.60 ± 15.58 μg/g creatinine, respectively. In contrast, the mean concentration of zinc protoporphyrin in the blood was 18.35 ± 22.00 μg/dL. Exceedances of allowable biological concentrations were observed in the urinary concentrations of arsenic of 5.2% of subjects, in the blood cadmium and lead of 1.3% of subjects, and the ZnPP of 2.6% of subjects. The toxicological characteristics of the studied group of people professionally exposed to metals are summarized in Table 2.

### 3.2. Neurological Examination and Cognition

The neurological examination was normal in 33 patients (80.5%). In 8 cases (19.5%), symptoms of peripheral nervous system involvement were present. Symptoms were attenuation of superficial sensation in the distal parts of the legs and decreased ankle and knee reflexes. The results of the MMSE, SDMT, and CDT tests were normal in all patients. None of the patients showed abnormalities in the brain CT or MRI studies.

### 3.3. Evoked Potentials

Abnormal VEP was recorded in 4 patients (9.75%). In all patients, there were unilateral prolonged P100 VEP latencies. Abnormal VEP did not occur in patients with symptoms of peripheral nervous system involvement. No abnormal BAEP was found in any of the patients.

The mean P100, N145, and P100 Lo-Po VEP latencies were significantly longer in the arsenic-, cadmium-, and lead-exposed group, compared to the control group (Table 3).

The mean wave III and V BAEP, and the III–V and I–V interpeak latencies were significantly longer in the arsenic-, cadmium-, and lead-exposed group than in the control group, and the I and V BAEP amplitudes were significantly lower (Table 4).

### 3.4. Correlation Analysis

The correlation analysis did not show the existence of statistically significant linear relationships between the basic toxicological parameters and the VEP and BEAP parameters.

### 3.5. Regression Analysis

In the next stage, univariate regression analyses were performed between the toxicological variables and the tested VEP and BEAP parameters. Statistically significant relationships between the fact of occupational exposure to As, Cd, and Pb and selected VEP and BAEP variables were demonstrated: positive in the context of P100, N145, III, V, III–V and IV, and negative in the context of amplitude I and amplitude V. Significant relationships between As-U, Cd-B, Pb-B, and ZnPP concentrations and the studied VEP and BAEP variables were not demonstrated in the group of people exposed to As, Cd and Pb (Table 5).

## 4. Discussion

Metals that are often used in industry are the main source of exposure to toxins for workers. For this reason, government agencies regulate acceptable exposure levels, which is important for the safety of employees. While the necessary metals play important physiological roles in the body, heavy metals pose a significant health risk in cases of acute and chronic exposure at high levels. The CNS is particularly sensitive to intoxication [4,7,11]. The brain easily accumulates metals that are physiologically incorporated into the metalloproteins necessary for normal neuronal processes and the maintenance of energy homeostasis. Serious consequences may result from an excess of necessary metals or exposure to heavy metals that are not relevant to the functions of the nervous system [5,18,19]. There are several features that make the CNS sensitive to xenobiotics. These features include the complex structure of the nervous system, its long period of development, a diverse array of post-mitotic cells, the process of myelination, interconnection with other organs and systems, selective transport to the CNS, and a high metabolic rate [5,20]. The exact mechanisms responsible for the toxic effects of heavy metals on the nervous system are still under investigation. Despite many years of research, dose–effect relationships and the exact anatomical location of the observed clinical effects are still unknown [3,4,7,11].

Under normal conditions, Cd rarely reaches the brain in adults, because it is protected by a highly selective blood-brain barrier (BBB). This xenobiotic can bypass the BBB to reach the CNS through the olfactory pathway, especially in cigarette smokers and those exposed to air pollution. Cd-dependent BBB changes have also been shown to increase its permeability [21]. The consequence of this process is intracellular metal accumulation, induction of neurotoxicity, and impairment of nerve tissue. In experimental studies, it was found that Cd exposure to pregnant rats can have a detrimental effect on offspring memory, which was explained by changes in the transcriptional state of genes and the activity of antioxidant enzymes [22].

Both experimental animal and human studies have shown that early exposure to lead is associated with reduced BBB integrity, altered synaptogenesis, myelin microstructure, and metabolic brain content [23]. Magnetic resonance imaging studies have found a reduced volume of some brain structures [24]. Recently, using fMRI in fetuses exposed to Pb was shown to disturb the functional organization of the human brain network [23]. Chronic exposure to arsenic during development induces behavioral deficits accompanied by higher levels of soluble and membranal receptors for advanced glycation end products (RAGE) [25]. This leads to the formation of neurodegenerative changes associated with amyloid accumulation. Perinatal exposure to arsenic has also been shown to cause neurochemical changes to structures critically involved in motor and cognitive functions in the striatum, the frontal cortex, and the hippocampus [26]. Experimental studies have demonstrated the harmful effects of Cd, As, and Pb on various animal organs and interactions between toxic elements, sometimes additive ones [7,27].

Evoked potential (EP) studies are a particularly useful electrophysiological method used in the diagnosis of functional CNS disorders. In contrast to imaging techniques, which reveal only anatomical and structural changes, EPs allow for the assessment of the integrity and functional activity of the nerve pathways. They are particularly helpful in detecting clinically silent disorders, locating lesions, confirming questionable and ambiguous changes, and monitoring the course of some neurological diseases. Previous reports on both exogenous and endogenous evoked potentials in patients with occupational exposure to heavy metals are few and often relate to casuistic cases of rare complications [1,9,28]. There are also no studies on groups of patients exposed to numerous heavy metals at the same time. Sińczuk-Walczak et al. conducted a VEP study on a group of 21 copper melting factory employees exposed to the toxic effects of arsenic. In 9 workers (42.8%) an incorrect VEP was registered. The authors showed a significantly longer P100 wave latency in the study group than in the control (101.20 vs. 96.20 ms, *p* = 0.01), there were no significant changes in the amplitude and relationship between VEP parameters and arsenic concentration in the urine of employees and the air [1]. Discalzi et al. analyzed the BAEP of 22 factory workers exposed to lead and mercury. Workers showed significant prolongation of the I–V BAEP wave latencies (4.04 vs. 3.89 ms, *p* = 0.001). Discalzi et al. also showed that in the case of Pb> 50 μg/dL in the blood, the I–V BAEP wave latency was longer than in patients with a concentration below 50 μg/dL (4.10 vs. 3.98 ms, *p* = 0.034) [29]. Other researchers did not confirm the effect of heavy metals on EP parameters and did not correlate with their levels [30,31]. Ejma et al. examined the central conduction time (TT) obtained by stimulation of the median nerves in 15 inhabitants of the area around a copper smelter (Legnica) exposed to industrial toxic substances (mainly heavy metal oxides) with slight toxic polyneuropathy. They revealed the difference in latency of two somatosensory evoked potential (SSEP) components recorded from the parietal cortex (N20) and the brainstem (N13). The TT was abnormal in 11 of 15 patients, and in the group of people with toxic polyneuropathy, it was significantly longer (*p* <0.001) [32].

Few experimental studies of animal heavy metal intoxication also suggest a significant effect on the CNS. Yargicogluo et al. analyzed the VEP of rats exposed to cadmium in prenatal life and after birth and showed extended latencies and reduced amplitudes of VEP components in rats exposed to cadmium compared to the control group [33]. In experimental studies of rats, cadmium ions were shown to increase lipid peroxide levels in rat brains and brain microvessels [34]. Shukla et al. [35] suggest that BBB damage caused by cadmium may result from an inhibition of the antioxidative defense system in brain microvessels. Agar et al. [36,37] showed a decrease in the amplitude in EEG recordings, prolongation of all SSEP components, and an increase in brain lipid peroxidation in female rats treated with cadmium compounds. Lilienthal et al. analyzed BAEP in monkeys exposed to lead toxicity. They showed an increased component latency of the BAEP in exposed monkeys compared to the control group [38].

In our patients, we found a significant increase in the N75 and P100 VEP latencies. Analyzing BAEP, we found a significant increase in the latency of all components and a significant decrease in the amplitude of the I and V waves. Our research shows that patients with occupational exposure to arsenic, cadmium, and lead show signs of brain bioelectrical dysfunction, manifested by the impairment of nerve conduction in the visual and auditory pathways.

Recent studies devote substantial attention to the significant and persistent harmful effects of heavy metals on developing organisms. However, our research subjects were exposed to Cd, As, and Pb only in adulthood. These people did not have symptoms of a central neurological deficit or visible cognitive impairment, and the concentration of toxic elements in the blood and urine in most employees did not exceed the acceptable biological concentrations (maximum admissible concentrations were exceeded only in approx. 5% of the subjects in the case of As-U, approx. 3% of the subjects in the case of ZnPP, and approx. 1% of the subjects in the case of Cd-B and Pb-B, see Table 2). Despite this, the study group showed brain bioelectrical dysfunction in the form of changes in EP latency and amplitudes. The obtained results indicate that occupational chronic exposure to metals disrupts CNS functions, even in the case of levels substantially within the permissible levels of exposure.

The EP test is a highly sensitive method that can be useful for recognizing and monitoring occupational exposure to heavy metals. It can also be easily and cheaply used in preventive examinations, allowing for the detection of early, small, and possibly reversible CNS disorders. The authors recommend regular preventive examinations of infants and children living in areas with exposure to toxic substances in the future.

The main limitations of the study were the small size of the study group and no measurements of metals concentration in the control group, as it was not possible to recruit people agreeing to determine the concentration of metals in the blood and urine.

## 5. Conclusions

Occupational exposure to As, Cd, and Pb is associated with prolonged latency and reduced evoked potential amplitude, but As-U, Pb-B, Cd-B, and ZnPP concentrations are not linearly related to potential components. An evoked potential analysis may be a useful method of assessment of the central nervous system in patients with occupational exposure to heavy metals.

## Figures and Tables

**Table 1 brainsci-11-00350-t001:** General characteristics of the study groups.

	Arsenic-, Cadmium-, and Lead-Exposed Group *n* = 41			Control Group *n* = 36			*p* Value
	Mean	Median	SD	Mean	Median	SD	
Age [years]	51.27	54.00	10.44	51.08	51.50	10.41	0.94
Height [m]	175.07	175.00	7.02	173.56	174.00	6.34	0.32
Weight [kg]	85.22	86.00	10.72	80.50	80.50	9.12	0.06
BMI [kg/m^2^]	27.63	27.70	2.97	26.68	27.11	1.81	0.11
	**Number**		**Percent**	**Number**		**Percent**	***p* Value**
Gender							
Male	40		97.6	35		97.2	0.91
Female	1		2.4	1		2.8	0.91
Overweight/Obesity	34		82.9	29		80.6	0.79

**Table 2 brainsci-11-00350-t002:** Basic toxicological parameters in the study population.

	Mean	Median	SD	Minimum	Maximum
As-U [μg/g]	15.60	11.30	15.58	2.47	88.00
Cd-B [μg/L]	1.19	0.48	1.40	0.01	7.17
Pb-B [μg/L]	181.85	128.00	139.88	8.56	610.00
ZnPP [μg/dL]	18.35	11.00	22.00	1.00	106.00
	**Number**		**Percent**		
As-U > MAC (>35 μg/L)	4		5.2		
Cd-B > MAC (>5 μg/L)	1		1.3		
Pb-B > MAC (>500 μg/L)	1		1.3		
ZnPP > MAC (>70 μg/dL)	2		2.6		

MAC: maximum admissible concentration.

**Table 3 brainsci-11-00350-t003:** Mean values of the latency (ms) and amplitude (µV) of visual evoked potential VEP parameters in the arsenic-, cadmium-, and lead-exposed group and in the control group.

VEP	Arsenic-, Cadmium-, and Lead-Exposed Group *n* = 41			Control Group *n* = 36			*p* Value
	Mean	Median	SD	Mean	Median	SD	
Latency (ms)							
N75	72.28	70.25	9.53	70.43	69.88	4.81	0.29
P100	107.21	105.50	7.10	100.63	100.00	4.31	<0.0001
N145	155.87	156.00	13.81	140.94	141.75	10.46	<0.0001
P100 Lo-Po	3.17	2.50	2.80	1.55	1.00	1.58	0.001
Amplitude (µV)							
P100/N145	10.04	11.45	4.57	9.67	9.68	3.13	0.68

**Table 4 brainsci-11-00350-t004:** Mean values of the latency (ms) and amplitude (µV) of brainstem auditory evoked potential (BAEP) parameters in arsenic-, cadmium-, and lead-exposed group patients and in the control group.

BAEP	Arsenic-, Cadmium-, and Lead- Exposed Group *n* = 41			Control Group *n* = 36			*p* Value
	Mean	Median	SD	Mean	Median	SD	
Latency (ms)							
I	1.70	1.67	0.14	1.65	1.63	0.13	0.16
III	3.89	3.86	0.17	3.79	3.79	0.13	0.001
V	5.95	5.93	0.25	5.66	5.68	0.17	<0.001
I-III	2.22	2.18	0.23	2.14	2.17	0.09	0.05
III-V	2.06	2.03	0.16	1.85	1.82	0.15	<0.001
I-V	4.23	4.28	0.26	4.00	3.99	0.15	<0.001
Amplitude (µV)							
I	0.21	0.19	0.12	0.30	0.30	0.11	<0.001
V	0.38	0.37	0.13	0.46	0.46	0.17	<0.001

**Table 5 brainsci-11-00350-t005:** Results of univariate regression analyses in the study group.

		As, Cd i Pb	As-U	Cd-B	Pb-B	ZnPP
VEP	N75	ns	ns	ns	ns	ns
	P100	Rc = 6.581 *p* = 0.002	ns	ns	ns	ns
	N145	Rc = 14.934 *p* < 0.001	ns	ns	ns	ns
	P100/N145 amplitude	ns	ns	ns	ns	ns
BAEP	I	ns	ns	ns	ns	ns
	III	Rc = 0.100 *p* = 0.006	ns	ns	ns	ns
	V	Rc = 0.293 *p* < 0.001	ns	ns	ns	ns
	I–III	ns	ns	ns	ns	ns
	III–V	Rc = 0.205 *p* < 0.001	ns	ns	ns	ns
	I–V	Rc = 0.232 *p* = 0.003	ns	ns	ns	ns
	I amplitude	Rc = −0.140 *p* < 0.001	ns	ns	ns	ns
	V amplitude	Rc = −0.170 *p* < 0.001	ns	ns	ns	ns

## Data Availability

The data presented in this study are available upon request from the corresponding author. The data are not publicly available.

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
