# Peer review of "The Relationship between Occupationally Exposed Arsenic, Cadmium and Lead and Brain Bioelectrical Activity—A Visual and Brainstem Auditory Evoked Potentials Study"

_brainsci, 2021, doi:10.3390/brainsci11030350_

Round 1

Reviewer 1 Report

The authors made a sincere effort which is reflected in the manuscript. I have only one question before the  this manuscript will be well done- How did the authors unify the baseline characteristics of the samples?

Author Response

Dear Reviewer,

Thank you for careful and thorough reading of this manuscript and for the thoughtful comments. We are incredibly grateful for the review.

Comment of Reviewer: The authors made a sincere effort which is reflected in the manuscript. I have only one question before this manuscript will be well done - How did the authors unify the baseline characteristics of the samples?

Changes carried out in the paper: For the performed laboratory analyzes, precision errors and coefficients of variation were determined. The precision errors were calculated according to the formula: "± 2 * standard deviation". The coefficients of variation were calculated according to the formula: "(standard deviation / mean) * 100%". The obtained values of precision errors and coefficients of variation were within the limits of laboratory standards.

Best regards,

Authors

Reviewer 2 Report

I appreciate the opportunity to review the manuscript:  „ The relationship between occupationally exposed to arsenic, cadmium and lead and brain bioelectrical activity – a visual and brainstem auditory evoked potentials study” Ref: Brain Sciences (ISSN 2076-3425).

Recommendation: Accept

General comments:

The manuscript presented for review is very interesting and raises a very important issue.

The aim of this study was to evaluate parameters of visual and brainstem auditory evoked potentials in patients with occupationally exposed to arsenic, cadmium and lead.

The study group comprised 41 workers of Copper Smelter and Rafinery (average age: 51.27) with occupational exposure to arsenic, cadmium and lead.

The control group consisted of 36 healthy volunteers (35 men and 1 woman, aged 27-66, average age: 51.08).

Neurological examination, brain imaging, visual and brainstem auditory evoked potentials were performed and the relationship between blood Cd, Pb concentration (Cd-B, Pb-B), blood zinc protoporphyrin (ZnPP) and urine As concentration (As-U) were measured.

Strong points of the paper: topicality of the subject, the social utility, especially because results of this study strongly indicated  that evoked potentials may be useful methods of assessment of the central nervous system in patients with occupational exposure to heavy metals.

The Title reflects well the content of manuscript.

The Abstract reflects the content of manuscript.

Introduction is well written and the aim is clearly formulated.

Material and Methods: The research methods are appropriate, modern and give reliable results. The methods and equipment were validated and the sensitivity of analysis was sufficient. The assumed statistical methods were appropriate and correctly used. Methodologically, the work is very good, I would like to warmly congratulate the authors of the work. However, the problem is the lack of determination of the concentration of elements in blood and urine in patients from the control group. However, the problem is the lack of determination of the concentration of elements in blood and urine in patients from the control group.

Jednak problemem jest brak oznaczenia stężenia pierwiastków we krwi i moczu u pacjentów z grupy kontrolnej.

problem is that no indication of the concentration of elements in the blood and urine of patients of the control group.

Problem polega na tym, że brak wskazań na stężenie pierwiastków we krwi i moczu pacjentów z grupy kontrolnej.

Nie mogę wczytać wszystkich wyników

Ponów próbę

Ponawianie próby

Ponawianie próby

The authors of the work marked this as the limit of the study.However, the authors of the work marked this as the limit of the study.

Jednak autorzy pracy zaznaczyli to jako granicę badań.

However, the authors have indicated this as the limit of the study.

Jednak autorzy wskazali to jako granicę badania.

Nie mogę wczytać wszystkich wyników

Ponów próbę

Ponawianie próby

Ponawianie próby

Results: Results are presented in a clear and comprehensive way.

Discussion: The literature reference is correctly cited and the discussion is clear. The conclusions are appropriate and the literature cited in the text corresponds to the References.

The paper is clear and well written and provides new information.

Author Response

Dear Reviewer,

Thank you for careful and thorough reading of this manuscript and for the thoughtful comments. We are incredibly grateful for the review.

Comment of Reviewer: I appreciate the opportunity to review the manuscript: „The relationship between occupationally exposed to arsenic, cadmium and lead and brain bioelectrical activity – a visual and brainstem auditory evoked potentials study” Ref: Brain Sciences (ISSN 2076-3425).

Recommendation: Accept

General comments: The manuscript presented for review is very interesting and raises a very important issue. The aim of this study was to evaluate parameters of visual and brainstem auditory evoked potentials in patients with occupationally exposed to arsenic, cadmium and lead. The study group comprised 41 workers of Copper Smelter and Rafinery (average age: 51.27) with occupational exposure to arsenic, cadmium and lead. The control group consisted of 36 healthy volunteers (35 men and 1 woman, aged 27-66, average age: 51.08). Neurological examination, brain imaging, visual and brainstem auditory evoked potentials were performed and the relationship between blood Cd, Pb concentration (Cd-B, Pb-B), blood zinc protoporphyrin (ZnPP) and urine As concentration (As-U) were measured.

Strong points of the paper: topicality of the subject, the social utility, especially because results of this study strongly indicated  that evoked potentials may be useful methods of assessment of the central nervous system in patients with occupational exposure to heavy metals.

The Title reflects well the content of manuscript.

The Abstract reflects the content of manuscript.

Introduction is well written and the aim is clearly formulated.

Material and Methods: The research methods are appropriate, modern and give reliable results. The methods and equipment were validated and the sensitivity of analysis was sufficient. The assumed statistical methods were appropriate and correctly used. Methodologically, the work is very good, I would like to warmly congratulate the authors of the work. However, the problem is the lack of determination of the concentration of elements in blood and urine in patients from the control group. However, the problem is the lack of determination of the concentration of elements in blood and urine in patients from the control group.

Problem is that no indication of the concentration of elements in the blood and urine of patients of the control group. The authors of the work marked this as the limit of the study.

Results: Results are presented in a clear and comprehensive way.

Discussion: The literature reference is correctly cited and the discussion is clear. The conclusions are appropriate and the literature cited in the text corresponds to the References.

The paper is clear and well written and provides new information.

Changes carried out in the paper: The reviewer does not indicate the need to amend the article. Thanks again for the positive review of the article.

Best regards,

Authors

This manuscript is a resubmission of an earlier submission. The following is a list of the peer review reports and author responses from that submission.

Round 1

Reviewer 1 Report

I have performed a plagiarism check by Turnitin and the work seems to have plagiarism. Please see the attached file as an example, rewrite, and check before submission.

Thank you 

Author Response

Responses to the Reviewer's suggestions are included in the attached file.

Reviewer 2 Report

In the abstract, the author wrote ' copper smelter and rafinery' is it supposed to be rafinery or refinery?

The second last paragraph of the introduction, the author wrote potentialsstudy as one word, please revise.

The authors must define the abbreviations used in the manuscript to make it easier for readers who are not in the health sector to understand. For instance, the author wrote that exposed patients underwent (CT and MRI), this was supposed to be explained further what these scans do. Before writing the abbreviation, it is best to write in full, e.g Computed Tomography (CT). revise throughout the manuscript.

The results section; is there any reason why the author decided to use different units for Cd, Pb and As (μg / l, μg / g, μg / dl). Unless there is a proper reason, all the units should be the same, this will be easier for the readers to see which of the 3 heavy metals in the area is in large quantities.

The discussion section ' the concentration of toxic elements in the blood and urine in most employees did not exceed the acceptable biological concentrations' what is the acceptable biological concentrations, compare the figures of the acceptable concentrations to those you found in your results section so that the readers can easily see the comparison.

The author must include future recommendations indicating whether someone should conduct the same study in infants who live in the affected areas or nearby to see if they are affected. What were the study limitations, was the 77 people the only ones that qualified or there were others that qualified and refused to be part of the study

The author should use only one referencing style than combine. Check the document.

Author Response

(The authors gave the same response as above.)

Reviewer 3 Report

The manuscript by Waliszewska-Prosol et al. describes a study to evaluate the parameters of visual and brainstem auditory potentials recorded from workers in a copper smelter. The study group of 41 workers was carefully matched with a control group with similar age, weight, and gender characteristics. The visual and brainstem auditory potentials observed in the study group were then compared to arsenic, cadmium, and lead levels that were measured in blood and urine. The authors also reported zinc protoporphyrin numbers in red blood cells from the study group.

Although the authors reported that they collected blood samples from the control group, no chemical data was reported for the control group. This is a serious flaw in the research design and should be corrected before the manuscript is accepted for publication.

The authors did not use concentration units consistently. The zinc protoporphyrin numbers were reported as µg/dl, blood cadmium and lead levels were reported as µg/l, and the arsenic urine level was reported as µg/g. The authors should use a consistent set of units throughout the manuscript, in the tables and in the text. The preferred unit of liquid measure for fluids withdrawn from humans is µg/dL. Also, the l in concentration units should be capitalized, i.e., L. The authors used a lowercase L throughout the manuscript.

The authors used graphite furnace atomic absorption spectroscopy to measure cadmium and lead blood concentrations. GFAAS is an appropriate technique for this matrix but blood samples present unique problems in analysis that must be overcome. The authors do not state if the blood samples were digested or placed into the instrument directly, without treatment. The authors do not state the conditions that were used and whether or not a background corrector was applied to correct for the smoke that would be generated from the organic matrix of the blood. The authors do not provide any quality control data and they do not list the source of reference materials that were used to calibrate the instrument before recording their data. These comments also apply to the determination of arsenic in urine using hydride generation atomic absorption spectroscopy.

The authors reported measuring zinc protoporphyrin in erythrocytes, but they did not describe how the red blood cells were separated from the whole blood sample. Furthermore, the authors do not explain the reason for reporting this measurement. Zinc protoporphyrin levels are often used to evaluate whether a patient suffers from lead poisoning. However, zinc protoporphyrin numbers are also affected by an iron-deficient diet. If the authors were using the zinc protoporphyrin numbers as an indicator of lead poisoning they should have also measured iron in the blood samples to eliminate the possibility of iron deficiency as a cause of any observed changes.

Author Response

(The authors gave the same response as above.)

Round 2

Reviewer 1 Report

Please check the plagarism. 

Reviewer 3 Report

Although the authors reported that they collected blood samples from the control group, no chemical data was reported for the control group. This is a serious flaw in the research design and should be corrected.

The authors did not use concentration units consistently. The zinc protoporphyrin numbers were reported as µg/dl, blood cadmium and lead levels were reported as µg/l, and the arsenic urine level was reported as µg/g. The authors should use a consistent set of units throughout the manuscript, in the tables and in the text. The preferred unit of liquid measure for fluids withdrawn from humans is µg/dL. Also, the l in concentration units should be capitalized, i.e., L. The authors used a lowercase L throughout the manuscript.

The authors used graphite furnace atomic absorption spectroscopy to measure cadmium and lead blood concentrations. GFAAS is an appropriate technique for this matrix but blood samples present unique problems in analysis that must be overcome. The authors do not state if the blood samples were digested or placed into the instrument directly, without treatment. The authors do not state the conditions that were used and whether or not a background corrector was applied to correct for the smoke that would be generated from the organic matrix of the blood. The authors do not provide any quality control data and they do not list the source of reference materials that were used to calibrate the instrument before recording their data. These comments also apply to the determination of arsenic in urine using hydride generation atomic absorption spectroscopy.

The authors reported measuring zinc protoporphyrin in erythrocytes, but they did not describe how the red blood cells were separated from the whole blood sample. Furthermore, the authors do not explain the reason for reporting this measurement. Zinc protoporphyrin levels are often used to evaluate whether a patient suffers from lead poisoning. However, zinc protoporphyrin numbers are also affected by an iron-deficient diet. If the authors were using the zinc protoporphyrin numbers as an indicator of lead poisoning they should have also measured iron in the blood samples to eliminate the possibility of iron deficiency as a cause of any observed changes.